# Gas-phase collision rate enhancement factors for acid-base clusters up to 2 nm in diameter from atomistic simulation and the interacting hard sphere model

Valtteri Tikkanen<sup>1</sup>, Huan Yang<sup>2</sup>, Hanna Vehkamäki<sup>1</sup>, and Bernhard Reischl<sup>1</sup>

<sup>1</sup>Institute for Atmospheric and Earth System Research / Physics, Faculty of Science, University of Helsinki, P. O. Box 64, FI-00014, Helsinki, Finland

Correspondence: Valtteri Tikkanen (valtteri.tikkanen@helsinki.fi)

**Abstract.** Collisions of neutral molecules and clusters is the most prevalent pathway in atmospheric new particle formation (NPF), with direct implications on air quality and climate. Until recently, these collisions have been modeled mainly using non-interacting hard sphere (NHS) models, which systematically underestimate collision and particle formation rates due to omission of long-range interactions. Lately, atomistic simulations which account for long-range interactions have been used to study neutral molecule-molecule and molecule-cluster collisions, but studies on cluster-cluster collisions have still been lacking despite the relevant role they can play e.g. in haze formation in polluted urban areas. We have therefore studied collisions between neutral clusters of N bisulphate and N dimethylammonium ions at T = 300 K up to N = 32 using atomistic molecular dynamics (MD) simulations. Direct simulation results have then been compared against both the traditional NHS model and the newly proposed interacting hard sphere (IHS) variant. We find the collision rates given by the NHS to be enhanced by a factor of 2.18-5.61 in the atomistic MD simulations, with enhancement decreasing with cluster size, and an asymptotic limit  $\approx 2$ . The IHS model yields a constant enhancement factor of 3.36 over the NHS model for collisions between same-sized clusters, which decreases with increasing cluster size ratio. Our results demonstrate how even collisions between clusters of tens of acid-base pairs at a relatively high temperature cannot be accurately modeled by neglecting long-range interactions. We also show that the MD results cannot be reproduced by simple point-particle models, highlighting the importance of atomistic details of intermolecular interactions.

#### 1 Introduction

Atmospheric aerosols affect Earth's radiative balance by scattering light and enhancing cloud formation since they act as condensation nuclei for cloud droplets (Seinfeld et al., 2016; Kulmala et al., 2013; Hallquist et al., 2009; Ramanathan et al., 2001). The quantity and physicochemical quality of aerosols in the air also have direct and often adverse effects on human health (Pope and Dockery, 2006; Pöschl, 2005). Most atmospheric aerosols are formed through sticking collisions between gas-phase molecules in the process called new particle formation (NPF) (Gordon et al., 2017; Vehkamäki and Riipinen, 2012). Most of these collisions happen between neutral partners, and a large portion of newly formed aerosols

<sup>&</sup>lt;sup>2</sup>Max Planck Institute for Chemistry, D-55128 Mainz, Germany

(i.e. secondary aerosols) are born starting from collisions between acid (such as sulfuric acid) and base (e.g. amines) molecules (Kirkby et al., 2016; Wagner et al., 2017). Such acid-base cluster collisions are important in supplying condensation cores for other atmospheric vapors (such as organic compounds) to grow on (Ehn et al., 2014).

Modeling of collisions of small particles in the atmosphere has traditionally relied on kinetic gas theory, which is a non-interacting hard sphere (NHS) model, where the colliding molecules/clusters are assumed to be non-interacting spheres with a well defined radius, typically calculated from bulk liquid density (Vehkamäki, 2006). Omitting long range interactions, such as the omnipresent Van der Waals interaction, the model systematically underestimates collision frequencies, as has been recently reported (Lehtipalo et al., 2016; Yang et al., 2018; Halonen et al., 2019; Neefjes et al., 2022; Yang et al., 2023). The degree of this deviation between the NHS model and real collision events is expected to depend on the conditions: first, at higher temperatures the discrepancy is expected to decrease, as the thermal energy of the collision partners increases compared to the interaction energy, however atmospheric temperatures may not be high enough to justify the omission of long range interactions. Second, increasing size of collision partners could also reduce the discrepancy, if the attractive interaction increases less with size than the momenta of the colliding clusters.

In recent years, computer simulations have been used to study molecule-molecule/cluster collisions. Yang et al. (2018) used molecular dynamics (MD) trajectory calculations to study the collision rate coefficient of metal clusters in higher temperature aerosol synthesis systems. Halonen et al. (2019) used similar trajectory simulations to calculate collision rate coefficients for two sulfuric acid molecules. In addition, they calculated collision rate coefficients using a central field (CF) model, where the attractive interaction was fitted to a potential of mean force (PMF) calculation, carried out using well-tempered metadynamics (Barducci et al., 2008). Both, trajectory MD and the central field model, yielded very similar results, showing enhancement by a factor of 2.2-2.7 over collision rates calculated by the non-interacting hard sphere model, matching experimental findings of Lehtipalo et al. (2016). These approaches have since been used to study ion-dipole (Neefjes et al., 2022) and neutral molecule-cluster collisions (Yang et al., 2023). The latter study also introduced a new theoretical framework, the interacting hard sphere (IHS) model.

In most atmospheric conditions, molecule-cluster collisions are far more frequent than cluster-cluster collisions. However, it is known that in certain cases, such as polluted urban air, the collision and merging of larger particles, process known as coagulation, has a significant effect e.g. on haze formation (Guo et al., 2014). Therefore, in this paper we focus on cluster-cluster collisions to see how well both the old and the new modeling approaches work when applied to these larger systems. We carry out and extensively analyze a set of new cluster-cluster trajectory MD simulations, and also extend the newly proposed interacting hard sphere model to cover collisions between clusters. MD results are compared to the IHS model and also against the traditional non-interacting hard sphere model. While the limitation to same-sized cluster collisions in the MD simulations in this work does not account for the vast majority of asymmetric collisions in real atmospheric processes, it still provides a useful starting point to investigate size-dependent collision rate enhancements, when attractive cluster-cluster interactions are taken into account.

The paper is organized as follows: in section 2, we present the theoretical and computational methods used in this study. In section 3 we present acid-base cluster properties and cluster-cluster collision rate coefficients from atomistic simulations, and compare to both the non-interacting and interacting hard sphere model results. Finally, in section 4 we discuss the results and present our conclusions.

# 2 Theoretical and computational methods

Traditionally, collisions in the gas phase have been modeled by using kinetic gas theory, in which the gas consists of rigid spherical particles interacting only through elastic collisions. The collision rate coefficient of such non-interacting hard spheres (from now on, NHS) is:

65 
$$\beta_{\text{NHS}} = \sqrt{\frac{8k_{\text{B}}T}{\pi\mu}}\pi \left(R_i + R_j\right)^2 = \sqrt{\frac{16k_{\text{B}}T}{\pi m}}\pi \left(2R_{\text{HS}}\right)^2,\tag{1}$$

where  $k_{\rm B}$  is the Boltzmann constant, T is the system temperature,  $\mu = m_i m_j / (m_i + m_j)$  the reduced mass of the colliding particles with masses  $m_i$  and  $m_j$ , and  $R_i$  and  $R_j$  are the hard sphere radii of the colliding particles, respectively. In this work,  $m_i = m_j \equiv m$  and  $R_i = R_j \equiv R_{\rm HS} = R_i \equiv R_{\rm HS}$ , as only collision between clusters of the same size are studied. In general, the perpendicular distance between two colliding particles at the start of the trajectory is called the impact parameter, b, and the maximum value of b which still allows for a collision to occur is the critical impact parameter,  $b_c$ . In the NHS model, a collision will happen only if the linear trajectories of the colliding particles have a minimum distance  $r_{\rm min} \leq 2R = b_{\rm c,HS} r_{\rm min} \leq 2R_{\rm HS} \equiv b_{\rm c,HS}$ , where  $b_{\rm c,HS}$  denotes the critical impact parameter in the NHS model.

#### 2.1 Central field model

Considering a possible collision between two point-like particles in a vacuum, Landau and Lifshitz (1976) reduced the geometry to a single-body problem in a central field:

$$U_{\text{eff}}(r) = U(r) + \frac{L^2}{2\mu r^2} = U(r) + \frac{\mu v_0^2 b^2}{2r^2},$$
(2)

where r is the distance between the colliding body and the center of the field,  $L = \mu v_0 b$  is the angular momentum,  $v_0$  is the initial velocity and b is the impact parameter. Notably, the introduction of the centrifugal term  $L^2/(2\mu r^2)$ , coupled with the conservation of angular momentum, leads to an energy barrier known as the centrifugal barrier, which need to be overcome for a collision  $(r \to 0)$  to occur. This leads to a condition which needs to hold for r > 0:

$$U_{\text{eff}} - \frac{\mu v_0^2}{2} \le 0. \tag{3}$$

In other words, the system needs a sufficient initial translational kinetic energy to overcome the centrifugal barrier. In terms of the impact parameter, the condition is:

$$b^2 \le r^2 \left( 1 - \frac{2U(r)}{\mu v_0^2} \right) \equiv \omega(r), \tag{4}$$

where  $\omega(r)$  is defined by Eq. 4 for convenience, following the work of Yang et al. (2023). The minimum of  $\omega(r)$  determines the critical impact parameter in the central field model:

$$b_{cCF}^2 = \omega(r_{\min}),\tag{5}$$

where  $r_{\min}$  is the location at which  $\omega(r)$  reaches its minimum. However, due to the point-like particle approximation inherent in the central field approach, whether this expression works depends on the attractive potential  $U(r) = -A(r/r_0)^a$ , where A,  $r_0$  and a are system-specific constants. As explained by Yang et al. (2023), Eq. 5 does not hold for an arbitrary interaction exponent a, as for example -2 < a < 0, corresponding to e.g. Coulombic interaction, where a = -1, leads to  $b_{c,CF} = 0$ . However, for a < -2 and moderate relative velocities, i.e. for conditions expected for neutral clusters colliding in the atmosphere, the CF model yields  $b_{c,CF} > 0$ , as  $\omega(r)$  has a single minimum at  $r = r_{\min} > 0$ :  $\omega(r_{\min}) > 0$ .

# 2.2 The interacting hard sphere model

The interacting hard sphere model (IHS) recently introduced by Yang et al. (2023), is built upon the central field model, but the collision criterion based on point particles collapsing into each other  $(r \rightarrow 0)$  is relaxed to a physically more intuitive condition: in the IHS model, a collision is thought to have happened if the center of mass distance between the colliding particles is at any point below the sum of the hard sphere radii. In contrast to the NHS model, a collision may occur even if the colliding partners have an impact parameter larger than the sum of their hard sphere radii, i.e.  $b_c > 2Rb_c > 2R_{HS}$ . The critical impact parameter  $b_c$  under the IHS model depends on the location  $r_{min}$ , discussed in Sec. 2.1. The two possible scenarios are: 1)  $r_{min} \le 2R_{HS}$ , i.e. the location of the minimum is smaller or equal compared to the sum of the hard sphere radii of the colliding particles, or 2)  $r_{min} > 2R_{HS}$ , if the location of the minimum exceeds the sum of the hard sphere radii. Corresponding critical impact parameters for these distinct scenarios are: 1)  $b_c = \omega_v(r_{min})$  and 2)  $b_c = \omega_v(2R)b_c = \omega_v(2R_{HS})$ . Knowing  $b_c$  allows for calculation of the collision cross section (CCS):

105 
$$\Omega(\nu_0) = 2\pi \int_0^{b_c(\nu_0)} \underline{bdb} \underline{bdb} = \pi b_c^2(\nu_0),$$
 (6)

and, finally, the collision rate coefficient as

115

$$\beta_{\text{IHS}} = \int_0^\infty \Omega(\nu_0) \nu_0 f(\nu_0) d\nu_0,\tag{7}$$

where  $f(v_0)$  is the Maxwell-Boltzmann velocity distribution. The critical step for utilizing the IHS model for non-trivial systems, such as the cluster-cluster systems studied here, for which the form of the pair potential U(r) is not analytically known, is to obtain the potential and the corresponding  $r_{\min}$ . Here, we use the umbrella sampling technique (Torrie and Valleau, 1977) (see Sec. 2.4) to solve the potential for the monomer-monomer system,  $U_{\min}$ , and expand the result to cover larger clusters by using the approach of Hamaker (1937), which outputs the potential for the cluster-cluster system  $U_{\text{cc}}$  using  $U_{\min}$  as input (see Sec. 2.2.1). In this work, the term 'monomer' refers to the acid-base dimer unit,  $[\text{HSO}_4^{-1} \cdot (\text{CH}_3)_2 \text{NH}_2^{+1}]_1$ , because this 'heterodimer' is the logical unit in modeling and simulating neutral bisulphate-dimethylammonic clusters.

## 2.2.1 Effective cluster-cluster potential from Hamaker approach

120

135

Monomer-monomer interactions in atmospheric clustering are commonly described using the attractive component of the van der Waals potential. The repulsive component of the potential becomes significant only when the colliding entities are very close to each other, at which point they have already been assumed to adhere and form a new cluster. Consequently, the repulsive component is disregarded within this framework. The attractive van der Waals potential is represented by the equation

$$U_{\rm mm}(r) = -4\epsilon \left(\frac{\sigma}{r}\right)^6,\tag{8}$$

Where  $\epsilon$  represents the depth of the potential well,  $\sigma$  is a characteristic length, and the subscript "mm" denotes the monomer-monomer interaction. Here, the parameters  $\epsilon$  and  $\sigma$  can be acquired either through fitting to the potential of mean force (PMF) between two dimer units,  $[HSO_4^{-}\cdot(CH_3)_2NH_2^{+}]_{1,\epsilon}$  computed from molecular dynamics (MD) simulations or directly extracted from literature sources, particularly when computational resources are limited. Assuming a cluster with constant density and pairwise monomer-monomer interactions within the system, neglecting many-body effects, an approximate monomer-cluster potential can be derived by integrating the monomer-monomer potential over the volume of the cluster. This derivation yields:

130 
$$U_{\text{mc}}(r) = \iiint_{V_{-}} U_{\text{mm}}(r)\rho_{c1}dV = -4\rho_{c1}\epsilon\sigma^{6} \int_{0}^{2\pi} d\theta \int_{0}^{\pi} d\phi \int_{0}^{R_{1}} \frac{\rho^{2}\sin(\phi)}{[r^{2} + \rho^{2} - 2\rho\cos(\phi)]^{3}} d\rho = -\frac{4n_{c1}\epsilon\sigma^{6}}{(r^{2} - R_{1}^{2})^{3}},$$
 (9)

where  $\rho_{c1}$  is the monomer number density,  $R_1$  is the radius of the cluster,  $V_{c1}$  is the volume of the cluster, and  $n_{c1}$  is the total number of monomers in the cluster. Equation 9 is strictly speaking applicable only to clusters comprising identical monomer types; however, its applicability can be extended to scenarios involving clusters composed of a mixture of different monomers (for details, see Yang et al. (2023)). Additionally, the approximate cluster-cluster potential can be derived by integrating the monomer-cluster potential across the volume of the other colliding cluster:

$$U_{cc}(r) = \iiint_{V_{cc}} U_{mc}(r)\rho_{c2} dV = -\frac{4\pi^{2}\rho_{c1}\rho_{c2}\epsilon\sigma^{6}}{3} \left\{ \frac{R_{1}R_{2}}{[r^{2} - (R_{1} + R_{2})^{2}]} + \frac{R_{1}R_{2}}{[r^{2} - (R_{1} - R_{2})^{2}]} + \frac{1}{2} \ln \left[ \frac{r^{2} - (R_{1} + R_{2})^{2}}{r^{2} - (R_{1} - R_{2})^{2}} \right] \right\}, \quad (10)$$

where  $\rho_{c2}$  is the monomer number density,  $R_2$  is the radius, and  $V_{c2}$  is the volume of the other cluster. Equation 10 represents the well-known solution for the Hamaker potential, which characterizes the van der Waals attraction between two homogeneous spherical clusters or nanoparticles (Hamaker, 1937).

#### 140 2.2.2 Determination of the critical impact parameter

The critical impact parameter  $b_c$  can be determined using the expression given by Eq. 5 once the interacting potential of the two colliding entities is specified. Our previous research has established an analytical solution for calculating monomer-cluster collision rate coefficients (Yang et al., 2023), employing the potential defined in Eq. 9. Extend-

ing this approach to compute cluster-cluster collision rate coefficients theoretically follows the same framework. However, due to the inherent intricacy of the cluster-cluster potential in Eq. 10, deriving an analytical solution for Eq. 5 becomes unfeasible. Consequently, we resort to numerical techniques to find first determine the minimum of the function  $\omega(r)\omega(r) = r^2(1-2U_{cc}(r)/\mu v_0^2)$  numerically. This numerical minimum is subsequently substituted into Eq. 5, facilitating the determination of the to determine the critical impact parameter. The obtained critical impact parameter,

$$b_{\rm c}^2 = \begin{cases} \omega_{\nu}(r_{\rm min}), & \text{if } r_{\rm min} > 2R_{\rm HS} \\ \omega_{\nu}(2R_{\rm HS}), & \text{if } r_{\rm min} \le 2R_{\rm HS} \end{cases},\tag{11}$$

50 is then used in Eqs. 6 and 7 to calculate the enhancement factor of the IHS model numerically.

#### 2.3 Atomistic models

To benchmark the non-interacting and interacting hard sphere model results, we carried out atomistic simulations of isolated  $[HSO_4^{-}\cdot(CH_3)_2NH_2^{+}]_N$  clusters for N=1...64, and performed collision trajectory simulations between same-sized clusters for N=1,2,4,8 and 32, using the LAMMPS molecular dynamics code (Plimpton, 1995). As in previous studies (Loukonen et al., 2010; Halonen et al., 2019; Yang et al., 2023), the atomistic interactions were described by the OPLS (optimized potentials for liquid simulations) all-atom force field (Jorgensen et al., 1996), in which the intramolecular interactions are given as a sum of harmonic bond potentials between covalently bonded atoms, harmonic angle potentials between atoms separated by two covalent bonds, and dihedral angle potentials between atoms separated by three covalent bonds:

160 
$$U_{\text{intra}}^{\text{OPLS}} = \sum_{i=1}^{N_{\text{bonds}}} \frac{k_i^b}{2} (r_i - r_i^0)^2 + \sum_{i=1}^{N_{\text{angles}}} \frac{k_i^\theta}{2} (\theta_j - \theta_j^0) + \sum_{k=1}^{N_{\text{dishedrals}}} \sum_{n=1}^{4} \frac{V_n}{2} [1 + \cos(n\phi^k - \phi_n^k)],$$
 (12)

where  $k_i^b$ ,  $r_i$ , and  $r_i^0$  are the force constant, instantaneous, and equilibrium length of bond i,  $k_j^\theta$ ,  $\theta_j$ , and  $\theta_j^0$  are the force constant, instantaneous, and equilibrium value of angle j, and  $V_n$ ,  $\phi_n^k$ , and  $\phi_k$  are the Fourier coefficients, phase angles, and instantaneous value of the dihedral angle k.

The intermolecular interactions, acting upon atoms i and j separated by more than three covalent bonds at distance  $r_{ij}$ , are given as a sum of Lennard-Jones and Coulombic terms:

$$U_{\text{inter}} = \sum_{i=1}^{N_1} \sum_{j=1}^{N_2} 4\epsilon_{ij} \left[ \left( \frac{\sigma_{ij}}{r_{ij}} \right)^{12} - \left( \frac{\sigma_{ij}}{r_{ij}} \right)^{6} \right] + \sum_{i=1}^{N_1} \sum_{j=1}^{N_2} \frac{q_j q_j}{4\pi\epsilon_0 r_{ij}},\tag{13}$$

where  $\epsilon_{ij}$  and  $\sigma_{ij}$  are the Lennard-Jones parameters,  $q_i$  and  $q_j$  the partial charges and  $\epsilon_0$  the vacuum permittivity.

The configurations of the three smallest clusters considered in this study (N = 1, 2 and 4) were taken from the quantum chemical (QM) cluster database of Elm et al. (2016). These minimum energy structures are characterized by a strong degree of symmetry, and complete proton transfer between the acids and the bases, stabilizing the clusters. Larger clusters (N = 8, 16, 32 and 64) were constructed by first condensing bisulphate and dimethylammonium ions from vapor,

followed by an equilibration in NVT (canonical ensemble) simulations using a Nosé-Hoover thermostat at T = 300 K and finally energy-minimizing the formed structures. During the numerous tests carried out for this study we found that the exact starting configuration for the simulations are not relevant: at 300 K the studied clusters are quite liquid-like, having energies clearly higher than their QM/energy-minimized counterparts. We note that the OPLS forcefield is non-reactive, i.e., the protonation state is determined when the model system is constructed and cannot change during the simulation. For clusters N > 4, we have also assumed complete proton transfer, i.e., clusters consisting only of bisulphate and dimethylammonium ions, which may be questionable. However, on in line with the QM minimum energy structures of the largest clusters available in the database (Elm, 2019). On the one hand this increases the stability of the clusters during collision trajectory simulations, and on the other hand, the exact charge distribution within the cluster becomes less important, as the size of the cluster increases. We also note that all clusters are dry, i.e., they do not contain any water molecules. Snapshots of the clusters of size N = 1, 2, 8 and 32 are shown in Figure 1.

**Figure 1.** Atomistic models of neutral bisulphate-dimethylammonium clusters  $[HSO_4^{-}\cdot (CH_3)_2NH_2^{+}]_N$  for N=1, 2, 8 and 32. Carbon is cyan, nitrogen is blue, oxygen is red, sulfur is yellow, and hydrogen is white. For N=1 and 2, hydrogen bonds are indicated by dashed red lines.

### 2.4 Trajectory simulations

The most intuitive approach to use molecular dynamics simulations (MD) to study collisions is to set two clusters on a possible collision course at a relative velocity v and impact parameter b, integrate the equations of motion, and check whether a collision occurs. Then repeat the simulation many times with different initial conditions to gain sufficient collision statistics. After a representative set of (v, b) pairs has been adequately sampled, the MD-based collision rate coefficient can be calculated as:

$$\beta_{\text{MD}} = \pi \int_0^\infty d\nu \int_0^\infty db^2 \nu f(\nu) P(\nu, b), \tag{14}$$

where f(v) is the relative velocity distribution and P(v,b) is the collision probability for a given (v,b) pair.

We started by equilibrating the colliding clusters individually for 50 ps (timestep 1 fs) beyond the potential cutoff of 300 Å with a Langevin thermostat. This thermostat was recently shown by Halonen et al. (2023) to be the optimal choice

for equilibrating isolated molecules or small clusters, in particular to achieve equipartitioning of energy over internal degrees of freedom, whereas other global thermostats, such as the Nosé-Hoover thermostat, will fail in this respect. The net angular and linear momentum of the whole system was removed during the equilibration, and other system properties, such as temperature and total energy, were monitored.

After the equilibration, the Langevin thermostat was turned off and the trajectory simulation was carried out under constant energy conditions, still using the timestep of 1 fs. First, the clusters were moved to a center of mass distance of 200 Å along the x direction from their equilibration positions. Then, the clusters were given an initial velocity of  $\pm v_x/2$ , where  $v_x$  is the target velocity, towards each other. The trajectory simulation was then continued until either a collision or a clear fly-by was detected. The criteria for counting a successful collision was based on the hard sphere radii  $R_{\rm HS}$  of the colliding clusters: if the center of mass distance  $r_{\rm com} r$  dropped below  $2R_{\rm HS} + R_{\rm b}$ , a collision was counted. Different buffer values  $R_{\rm b}$  were tested, but the results were not sensitive to this choice, and a value of  $R_{\rm b} = 0.2R_{\rm HS}$  was ultimately used. In fact, for typical collisions  $r 

Figure 2. Schematic of the setup for collision trajectory molecular dynamics simulations with a relative velocity v (along the x direction) and impact parameter b (along the z direction). After equilibration with a Langevin thermostat beyond the cutoff of the atomistic interactions, the two clusters are moved within the range of interactions and given an initial velocity of  $\pm v_x/2$  to set them on a potential collision course. The center of mass distance between the clusters,  $r_{com}r_z$  is shown in red and the hard sphere radii of the clusters,  $r_{com}r_z$  are indicated by dashed blue circles around the centers of mass of the clusters.

# 210 2.5 Potential of mean force from umbrella sampling simulations

200

205

To compare the IHS model against the direct trajectory simulations, we need the monomer-monomer interaction potential to calculate the cluster-cluster interaction potential, following the Hamaker approach, as explained in Sec. 2.2.1. This requires good estimates for the monomer-monomer potential energy well depth  $\epsilon$  and the location  $\sigma$  at which U(r) = 0. To gain these estimates, we have used umbrella sampling simulations -(Torrie and Valleau, 1977) to construct the Helmholtz

free energy profile F(r), as a function of the center of mass distance between two  $[HSO_4^{-}\cdot(CH_3)_2NH_2^{+}]$  monomers, r, at T=300 K. We used the PLUMED plug-in (Tribello et al., 2014) for LAMMPS (Plimpton, 1995; Thompson et al., 2022) to carry out the simulations. Harmonic umbrella potentials with a spring constant of k=1.5 eV/Å $^2$  were used to constrain the center of mass distance r between 3 and 30 Å, in 0.5 Å intervals, to ensure efficient sampling and adequate overlap between the subsequent windows. The radii of gyration of each  $[HSO_4^{-}\cdot(CH_3)_2NH_2^{+}]$  monomer were constrained by a harmonic upper wall with a spring constant of k=10 eV/Å $^2$ , starting at the value of  $r_8=2.5$  Å $r_8=2.5$  Å, to ensure the dimer units remain intact at intermediate distances, while still allowing for necessary rearrangements upon cluster formation. A simulation timestep of 1 fs and potential cutoff radius of 60 Å for both Lennard-Jones and electrostatic interactions were used for all systems, and the temperature was controlled with a stochastic velocity rescaling (CSVR) thermostat (Bussi et al., 2007). Simulations were ran for 50 ns for sufficient averaging of different configurations. The free energy profile over the full range of center of mass distances was calculated from the biased probability distributions of center of mass distances in each window using the weighted histogram averaging method (WHAM) (Grossfield, 2024) with a bin width of 0.2 Å. To obtain the potential of mean force (PMF), U(r), we removed the configuration entropy term from the free energy profile,

$$U(r) = F(r) + 2k_{\rm B}T\ln(r). \tag{15}$$

The IHS parameters  $\epsilon$ , and  $\sigma = r_0/2^{1/6}$ , were fitted to the well depth of the minimum in the PMF, and the position of the minimum,  $r_0$ , respectively.

To assess the effective attractive interactions between larger clusters obtained through the Hamaker approach, based on the monomer-monomer interactions, we also calculated the atomistic potentials of mean force between clusters of sizes N = 2.8, and 32 using a similar approach. For these simulations, harmonic upper walls were put on the value of the radii of gyration at distances deemed appropriate to achieve a compromise between avoiding elongation of the clusters leading to coalescence already at intermediate distances, and too rigidly constraining the clusters to their original spherical geometries.

### 3 Results

# 3.1 Dipole moments

For collisions between neutral clusters, the collision rate coefficient can be significantly increased over the kinetic gas theory value if the clusters possess a large dipole moment. Orientationally averaged dipole-dipole Keesom interactions are of the form

$$U_{\text{dipole-dipole}}(r) = -\frac{\mu_1^2 \mu_2^2}{3(4\pi\varepsilon_0)^2 k_{\rm B} T r^6},\tag{16}$$

where  $\mu_i$  denotes the dipole moment of dipole i and  $\varepsilon_0$  the vacuum permittivity. Clusters containing N = 1, 2, 4, 8, 32, 6445  $[HSO_4^- \cdot (CH_3)_2 NH_2^+]$  monomers were simulated in isolation for at least 20 ns under NVT (canonical ensemble, with the Nosé-Hoover thermostat) conditions. The average instantaneous dipole moment  $\langle \mu \rangle$  and its standard deviation were calculated, and the results are listed in Table 1 and shown in Figure 3. The heterodimer 'monomer' (N=1) has a large average dipole moment of 13.4 Debye and by far the strongest dipole moment per number of constituent ion pairs. For the N=2 cluster, the average dipole moment drops to 3.0 Debye, comparable to the dipole moment of a neutral sulfuric acid molecule. For the larger clusters studied here, we observe an increase of the dipole moment with size, where the N=32 cluster exhibits a dipole moment comparable to the N=1 monomer. The standard deviations of the dipole moments also increase with cluster size as larger clusters are more 'liquid-like' and can explore more configurations with different dipole moment values.

**Figure 3.** Time average of the instantaneous dipole moment  $\langle \mu \rangle$ , and the corresponding sample standard deviation shown as vertical lines, for different  $[HSO_4^- \cdot (CH_3)_2 NH_2^+]_N$  cluster sizes N.

#### 3.2 Potentials of mean force

The free energy profiles and potentials of mean force (PMF) as a function of the center of mass distance between two  $[HSO_4^{-}\cdot(CH_3)_2NH_2^{+}]_N$  clusters (N=1,2,8 and 32) from the umbrella sampling simulations at T=300 K are shown in Figure 4. For N=1, the PMF exhibits a global minimum at r=4.05 Å with a well depth of 1.53 eV, and the attractive tail for r>8 Å is in excellent agreement with the rotationally-averaged dipole-dipole interaction (Eq. 16) using the average dipole moment from simulation (see Table 1), indicating that the long-range attractive interaction between the heterodimers can be effectively modeled as that between two point dipoles. However, for any larger clusters this is not true, highlighting the importance of atomistic details of intermolecular interactions.

The naive Lennard-Jones fit to the PMF, based solely on the position and depth of the global minimum as suggested by Yang et al. (2023), yields a very poor result for the attractive tail of the PMF. This can be explained by the presence of

| N  | $\langle \mu \rangle$ (Debye) | $r_{HS} R_{HS}$ (Å) | $r_{g} R_{g}$ (Å) | $n_{\mathrm{traj}}$ | $\beta_{\mathrm{NHS}}~(\mathrm{m}^3\mathrm{s}^{-1})$ | $\beta_{\mathrm{MD}}~(\mathrm{m}^3\mathrm{s}^{-1})$ | $\beta_{\rm IHS}~({ m m}^3{ m s}^{-1})$ | $W_{ m MD}$ | $W_{\mathrm{IHS}}$ |
|----|-------------------------------|---------------------|-------------------|---------------------|------------------------------------------------------|-----------------------------------------------------|-----------------------------------------|-------------|--------------------|
| 1  | $13.4 \pm 0.7$                | 3.64                | 2.10              | 500                 | $4.96 \times 10^{-16}$                               | $2.78 \times 10^{-15}$                              | $1.67 \times 10^{-15}$                  | 5.61        | 3.36               |
| 2  | $3.0 \pm 1.4$                 | 4.59                | 2.97              | 500                 | $5.58 \times 10^{-16}$                               | $1.42 \times 10^{-15}$                              | $1.87 \times 10^{-15}$                  | 2.54        | 3.36               |
| 4  | $5.8 \pm 2.7$                 | 5.78                | 3.85              | 200                 | $6.26 \times 10^{-16}$                               | $1.46 \times 10^{-15}$                              | $2.10 \times 10^{-15}$                  | 2.34        | 3.36               |
| 8  | $7.7 \pm 3.2$                 | 7.28                | 5.09              | 200                 | $7.02 \times 10^{-16}$                               | $1.61 \times 10^{-15}$                              | $2.36 \times 10^{-15}$                  | 2.29        | 3.36               |
| 16 | $10.2 \pm 4.3$                | 9.17                | 6.44              | -                   | $7.87 \times 10^{-16}$                               | -                                                   | $2.64 \times 10^{-15}$                  | -           | 3.36               |
| 32 | $13.8 \pm 5.9$                | 11.56               | 8.11              | 100                 | $8.85 \times 10^{-16}$                               | $1.93 \times 10^{-15}$                              | $2.97 \times 10^{-15}$                  | 2.18        | 3.36               |
| 64 | $19.0 \pm 8.0$                | 14.56               | 10.35             | -                   | $9.93 \times 10^{-16}$                               | -                                                   | $3.34 \times 10^{-15}$                  | -           | 3.36               |

Table 1. Combined results for  $[HSO_4^{-}\cdot (CH_3)_2NH_2^{+}]_N$  cluster properties and collisions between two same-sized  $[HSO_4^{-}\cdot (CH_3)_2NH_2^{+}]_N$  clusters: average dipole moment  $\langle \mu \rangle$  from simulation, hard sphere radius  $r_{HS}R_{HS}$ , radius of gyration  $r_gR_g$  from simulation, number of collision trajectories simulated  $n_{traj}$ , non-interacting hard sphere model collision rate coefficient  $\beta_{NHS}$ , MD trajectory simulation collision rate coefficient  $\beta_{IHS}$ , and the enhancement factors over the NHS model calculated based on the trajectory simulations  $W_{MD}$  and the interacting hard sphere model  $W_{IHS}$ . Bootstrap-based error estimates were calculated for the trajectory simulation results, yielding a minor inner uncertainty of  $\sim 0.01$  for the enhancement factors  $W_{MD}$  for all studied systems.

at least two shallow minima or shoulders in the PMF at much larger distances than the global minimum at r = 4.05 Å, corresponding to the distances at which the two clusters can first start to form hydrogen bonds at particular orientations. For clusters N = 2.8 and 32, the PMF curves shown in Figure 4 differ considerably from the monomer case. Whereas the dipole-dipole interaction fits the tail of the PMF almost perfectly for the monomer, for the larger clusters the contribution from the dipole-dipole interaction stays effectively at zero for center of mass distances  $> 2R_{\rm HS}$ , where the fate of the collision is determined. On the other hand, compared to the monomer system, the Hamaker approach (Eq. 10) is in much better agreement with the tails of PMF's for the larger clusters, explaining the smaller discrepancy between the collision rate coefficients  $\beta_{MD}$  and  $\beta_{IHS}$  shown in Table 1. It must be noted that for systems N > 1 we solely focus on the tails of the PMF curves, i.e. distances at which the PMF increases monotonically to zero, for multiple reasons: (1) the location and depth of the possible global minimum in the PMF is not needed in the Hamaker approach apart from the monomer (see Eq. 10). (2) The tail region of the PMF is what determines whether a collision will take place. (3) We faced technical difficulties in determining the PMF's for the larger clusters at close distances due to clusters merging into each other. Preventing this would require adding artificial constraints on the cluster geometries, which would be both unnecessary and could slightly affect the more important tail region of the PMF. Therefore, we omitted close-distance values from the PMF and free energy curves shown in Figure 4 for systems larger than N=2, and even the N=2 PMF might already suffer from artifacts near the global minimum.

**Figure 4.** Free energy profiles (blue) and potentials of mean force (red) from umbrella sampling between two  $[HSO_4^- \cdot (CH_3)_2 NH_2^+]_N$  clusters for N = 1, 2, 8 and 32. The dashed black line indicates the fit to the attractive tail of the PMF used for the Hamaker approach in the IHS model (N = 1), or the effective cluster-cluster Hamaker potential (N = 2, 8 and 32). The dot-dashed dark grey line represents the rotationally-averaged dipole-dipole interaction using the average dipole moment from simulation. The dashed light grey line indicates the distance equal to the sum of hard sphere radii,  $r = 2R_{HS}$ .

# 3.3 Trajectory simulations

280

We performed  $n_{\text{traj}} = 500$ , 500, 200, 200 and 100 individual MD trajectory simulations for collisions between N = 1, 2, 4, 8 and 32  $[\text{HSO}_4^{-} \cdot (\text{CH}_3)_2 \text{NH}_2^{+}]_N$  clusters, respectively, to balance between accuracy and increasing computational cost for larger system sizes. To ensure adequate sampling of the (v, b) plane for calculating collision probability histograms,

different bin widths for both, initial velocity and impact parameter, were tested. While the trajectory simulation results were not sensitive to the step size in velocity, dv, the step size for sampling impact parameters, e.g., every 4 Å instead of 2 Å, had a noticeable effect. Thus, we calculated collision rate coefficients using different bin widths, db, as shown in Figure 5, yielding a linear behavior between the bin width and the collision rate coefficient. Ultimately, we chose to report the extrapolated 'infinite sampling' value  $\beta_{\rm MD}(db \rightarrow 0)$ , for which there is also no difference between using either the central or upper value for each bin to calculate the collision frequency. Figure 5 illustrates how these two approaches converge at the infinite sampling limit, whereas using a finite bin width yields a higher, and increasing  $(\frac{\partial \beta_{\rm MD}}{\partial db} > 0) \frac{\partial \beta_{\rm MD}}{\partial db} > 0)$ , collision rate coefficient for the upper limit, when comparing against the value calculated using centered bins (for which  $\frac{\partial \beta_{\rm MD}}{\partial db} 

**Figure 5.** Collision rate coefficient  $\beta$  as a function of impact parameter histogram bin width for different cluster sizes  $[HSO_4^{-}\cdot(CH_3)_2NH_2^{+}]_N$ . Solid lines show the linear fits to the data for N=2, 8 and 32 (marked by circles, diamonds and squares, respectively) created by using centralized histogram bins. Dashed lines show the fits made to the results collected for the same systems by using the upper limit value for each histogram bin.

The collision probabilities as a function of relative velocity and impact parameter, P(v,b), are shown in Figure 6 for cluster sizes N=1,2,8 and 32. The corresponding collision rate enhancement factors  $W_{\rm MD}$  for all cluster sizes are shown in Table 1 and Figure 7. For the N=1 collision we obtain a very substantial enhancement factor of 5.61. As the sizes of colliding clusters increase, the enhancement factor decreases rapidly at first and reaches an asymptotic value around 2. Error estimates for the enhancement factors were calculated as the standard deviation of the bootstrapped sampling distribution. Only a minor uncertainty of  $\sim 0.01$  was found for the enhancement factor, implying adequate statistics, i.e., a sufficient amount of collision trajectories.

Details of individual MD trajectories for impact parameters and relative velocities of b = 16 Å and  $v = 384 \text{ ms}^{-1}$ , for N = 1, and b = 34 Å and  $v = 64 \text{ ms}^{-1}$ , for N = 32, are shown in Figure 8. For all systems, the distinction between collision

**Figure 6.** Heat maps of the collision probabilities P(v,b) as a function of relative velocity v and impact parameter b for collisions of a) N=1, b) N=2, c) N=8, and d) N=32 [HSO<sub>4</sub> $^-\cdot$  (CH<sub>3</sub>)<sub>2</sub>NH<sub>2</sub> $^+$ ]<sub>N</sub> clusters, from collision MD trajectory simulations. The orange curves overlaid on the heat maps show the Maxwell-Boltzmann velocity distribution, f(v), at T=300 K for each system. The cyan curves indicate the critical impact parameter,  $\frac{b_c(v)}{b_c(v)}$ , from the IHS model  $\frac{b_c(v)}{b_c(v)}$ 

, from the IHS model.

and fly-by is unambiguous: upon a successful collision, the clusters form a strongly bound complex that will not redissociate on the time scale of the simulation, even without any form of thermostatting that would dissipate the excess energy, as shown in Fig. 8a. In some cases the decrease in center of mass distance r is not monotonous, but after a first collision, it takes a few rotations of the clusters to explore different conformations before strong hydrogen bonds can be formed. However, for N = 1 the final values of r are all very similar and close to the sum of the radii of gyration  $r_g R_g$  of the isolated clusters. With increasing cluster size, the spread of r values of the formed complexes becomes larger, and the values are significantly larger than  $2r_g 2R_g$ , but still below the sum of hard sphere radii,  $2r_{HS} 2R_{HS}$ . Indeed, for the data presented here, the sum of hard sphere radii can be used as an easy collision criterion. Upon collision, hydrogen bonds are formed between acids and bases on the two clusters' surfaces, leading to a drop in potential energy, shown in Fig. 8b. For N = 1, we observe 'binding energies' of 1.0-1.3 eV, a bit less than the well-depth of 1.53 eV observed in the PMF calculation. This is expected, as the clusters are not equilibrated after the collision. For N = 32, the fluctuations in

Figure 7. Collision rate enhancement factor W for eluster-cluster collisions between two  $[HSO_4^- \cdot (CH_3)_2NH_2^+]_N$  clusters as a function of cluster size N from atomistic collision trajectory simulations (MD, shown as red squares ) and the interacting hard sphere model (IHS, shown as the dashed red line).

and the interacting hard sphere model (IHS, blue line), relative to non-interacting hard sphere collisions (NHS, grey line).

the potential energy are much larger, but the initial 'binding energy' appears to be of the same order of magnitude as for N=1, despite the larger contact area between the two clusters allowing for more hydrogen bond formations. Finally, it is interesting to note that the average dipole moments of the clusters also increase significantly upon collision, as shown in Fig. 8c.

# 3.4 Interacting hard sphere model for cluster-cluster systems

325

Based on the naive Lennard-Jones fit to the potential of mean force between two  $[HSO_4^{-}\cdot(CH_3)_2NH_2^{+}]_1$  clusters, we derived the effective attractive cluster-cluster interaction potentials between larger acid-base clusters of equal size,  $[HSO_4^{-}\cdot(CH_3)_2NH_2^{+}]_N$ , using Eqs. 9 and 10. These effective interactions between larger clusters are shown in Figure 4. We determined the interacting hard sphere model critical impact parameters  $b_c$ , shown in Figure 6 and collision rate enhancement factors  $\beta_{IHS}$ , reported in Table. 1, using Eq. 7. As shown in section 3.2, the naive fit to the monomer-monomer PMF underestimated the attractive interaction, which is reflected in the IHS model critical impact parameter curve  $b_c(v)$  not agreeing with the collision probability map from collision MD trajectory simulations for N=1 collisions, and a significantly lower collision rate enhancement factor,  $W_{IHS}=3.36$ , compared to  $W_{MD}=5.61W_{MD}=5.61$ . The agreement between the effective attractive cluster-cluster interaction potentials and the PMFs calculated from atomistic simulations becomes somewhat better for larger cluster sizes, as shown in Figure 4, and correspondingly the critical impact parameter curves match better with the collision probability heat maps obtained from collision MD trajectories shown in Figure 6.

Figure 8. Collision MD simulation results for two [HSO<sub>4</sub><sup>-</sup>·(CH<sub>3</sub>)<sub>2</sub>NH<sub>2</sub><sup>+</sup>]<sub>N</sub> clusters for N=1 and N=32: (a) Cluster-cluster center of mass distances r as a function of time for 20 independent trajectories (thin gray lines), at impact parameters and relative velocities of b=16 Å and v=384 ms<sup>-1</sup> (N=1), and b=34 Å and v=64 ms<sup>-1</sup> (N=32), leading to a collision probability  $P(v,b) \sim 0.5$  (see Fig. 6). For both cluster sizes, we highlight two trajectories leading to a collision (dark and light blue curves) and two trajectories corresponding to a fly-by (red and orange curves). The full and dotted black lines correspond to the sums of the clusters' hard sphere radii,  $r_{HS}R_{HS}$ , and radii of gyration,  $r_{g}R_{g}$ , respectively (see Tab.1). (b) Evolution of the potential energy  $U_{pot}$ , relative to the average value at the start of the simulation, indicated by the black line. (c) Evolution of the average instantaneous dipole moment  $\mu$  of the two clusters. The time-averaged dipole moments of the isolated clusters are indicated by the black line (see Tab.1). The colors of the curves in panels (b) and (c) correspond to the trajectories highlighted in panel (a).

However, the IHS collision rate enhancement factor,  $W_{\rm IHS}$ , does not change with increasing cluster sizes, increasingly over-predicting the values from atomistic simulation, starting with a  $\sim 30\%$  over-prediction at N=2.

330

The collision rate enhancement factor  $W_{IHS}$  remaining constant for all sizes of colliding cluster pairs studied was a surprising result. To gain further insights, we non-dimensionalized the equation of motion for cluster-cluster collisions in the IHS model (see Appendix A). In the IHS model, the enhancement factor depends only on the size ratio of collision

pairs, as shown in Figure 9 for collisions between clusters of different sizes between N = 1 and 131072. The maximum collision rate enhancement of 3.36 is obtained for collisions of equal-sized clusters. As the ratio of cluster sizes deviates from 1, the collision rate enhancement factors decrease, as was previously observed when the IHS model was applied to collisions between neutral acid or base molecules and acid-base clusters (Yang et al., 2023). For the largest cluster size ratio considered here (131072:1), the IHS enhancement factor drops to 1.3.

**Figure 9.** (a) Collision enhancement factor  $W_{\text{IHS}}$  shown as a heat map for collisions of different sized acid-base clusters  $[\text{HSO}_4^- \cdot (\text{CH}_3)_2 \text{NH}_2^+]_{\text{NL}}$  using the interacting hard sphere model (IHS) with cluster-cluster interactions determined from a Hamaker approach using the fitted monomer-monomer interaction. (b) IHS enhancement factor as a function of the cluster size ratio of colliding clusters.

#### 4 Discussion and conclusions

We have carried out atomistic molecular dynamics simulations to compute the average dipole moments and radii of gyration of neutral bisulphate-dimethylammonium clusters  $[HSO_4^{-}\cdot(CH_3)_2NH_2^{+}]_N$  up to N=64, as well as potentials of mean force between two same-sized clusters, and trajectory simulations for collisions of same-sized clusters of sizes N=1,2,4,8 and 32. The N=1 cluster exhibits a large average dipole moment, which drops significantly for N=2 and then steadily increases again up to N=64, the largest cluster studied. The potential of mean force (PMF) between two N=1 clusters is characterized by a deep global minimum and several shoulders along the attractive tail, caused by the formation of hydrogen bonds between the constituent ions. The long-range attractive tail is in excellent agreement with the rotationally averaged dipole-dipole interaction using the averaged dipole moment of the N=1 cluster. A naive Lennard-Jones fit, based on the position and depth of the global minimum ( $r_0=4.05$  Å and  $\epsilon=1.53$  eV) was used to determine an effective attractive Lennard-Jones interaction between 'monomers' to derive the effective attractive cluster-cluster interactions in the interacting hard sphere model (IHS) using a Hamaker approach. For the larger clusters, the

PMF calculations were complicated by the tendency of the clusters to elongate and coalesce at center of mass distances well above the sum of their initial radii of gyration.

The atomistic collision trajectory simulations carried out for cluster sizes N=1,2,4,8,32 exhibit size-dependent collision rate enhancement factors over non-interacting hard sphere model of kinetic theory. For N=1, the enhancement factor  $W_{\rm MD}=5.61$  is largest, and decays monotonically with increasing cluster sizes, to  $W_{\rm MD}=2.18$  for N=32, with an asymptotic limit around  $W_{\rm MD}\approx 2$ . The large enhancement factor for N=1 can be explained by the large dipole moment of the cluster compared to its size, but the size-dependent decrease of  $W_{\rm MD}$  can not be explained by a simple relationship between cluster properties such as dipole moment and radius. In fact, the potentials of mean force obtained for larger clusters show that the attractive tail of the interaction does not resemble a dipole-dipole interaction.

360

365

In contrast, the IHS model predicts a constant enhancement factor for all cluster sizes, as long as the colliding clusters are of the same size. Using the naive fit to the potential of mean force between two N=1 clusters, we obtain  $W_{\rm IHS}=3.36$ . This underpredicts the case of N=1, and overpredicts N>1 collisions. A better fit to the N=1 potential of mean force would lead to a better agreement in the enhancement factor for N=1, but worsen the agreement for all larger clusters. This illustrates the limitations of the simple Hamaker approach used in the IHS framework for estimating the attractive interactions of larger clusters based on the interactions between their 'monomers', when the clusters are more complicated than an ideal Lennard-Jones system, as the atomistic potentials of mean force clearly indicate. The IHS approach has shown to work well for molecule-cluster collisions studied in earlier work by Yang et al. (2023). Despite its obvious shortcomings, the IHS model based on the naive Lennard-Jones fit is still able to predict an asymptotic enhancement factor value within a 50 % error margin of the atomistic simulation benchmark, offering at least a modest improvement over kinetic theory, for a quite complex system, at a cheap computational cost. However, if accurate collision rate coefficients are needed for complex systems, the present study clearly indicates that atomistic collision trajectory simulations with proper sampling of the relevant ranges of impact parameters and relative velocities are still needed.

The substantial collision rate enhancement factor due to long-range interactions found for clusters containing a single acid-base pair is certainly relevant for atmospheric new particle formation. However, for collisions between larger, same-sized clusters, the enhancement factors quickly drop below 3, which is comparable in magnitude to the collision rate enhancement of "monomers" (monomers", such as two sulfuric acid molecules, obtained from similar atomistic MD simulations (Halonen et al., 2019). To quantitatively assess the effect of enhancement factors for collisions of molecules and/or clusters on the actual particle formation rates in polluted environments, the complete cluster size distribution dynamics (Mcgrath et al., 2012) would need to be simulated for the different growth pathways, using either the standard rate coefficients from kinetic gas theory, or those obtained from atomistic simulations, for all relevant collisions.

*Data availability.* All data needed to evaluate the conclusions in the paper are present in the paper. Additional simulation data and/or simulation input files can be made available by the authors upon reasonable request.

# Appendix A: Dimensionless form of the IHS model

The equation of motion for the clusters is:

$$m_{ij}\frac{d^2\mathbf{r}}{dt^2} = -\frac{\mathbf{r}}{r} \cdot \frac{dU_{cc}}{dr},\tag{A1}$$

where

$$U_{cc}(r) = -\frac{4\pi^2 \rho_1 \rho_2 \epsilon \sigma^6}{3} \left( \frac{R_1 R_2}{(r^2 - (R_1 + R_2)^2)} + \frac{R_1 R_2}{(r^2 - (R_1 - R_2)^2)} + \frac{1}{2} \ln \left[ \frac{r^2 - (R_1 + R_2)^2}{r^2 - (R_1 - R_2)^2} \right] \right). \tag{A2}$$

The collision rate enhancement factor is:

$$W = \frac{\int_0^\infty \pi b_c^2 v f(v) dv}{\pi (R_1 + R_2)^2 v_0},$$
(A3)

where

$$f(v)dv = \left(\frac{m_{ij}}{2\pi kT}\right)^{3/2} 4\pi v^2 \exp\left(-\frac{m_{ij}v^2}{2kT}\right) dv,\tag{A4}$$

and

$$v_0 = \left(\frac{8kT}{\pi m_{ij}}\right)^{1/2}.\tag{A5}$$

We non-dimensionalize the above equations using

$$r = R_1 r^*, \quad v = v_0 v^*, \quad t = \frac{R_1}{v_0} t^*, \quad U_{cc} = \frac{8kT}{\pi} U_{cc}^*.$$
 (A6)

We obtain the following non-dimensionalized form of the equation of motion:

$$m_{ij}\frac{d^2\mathbf{r}^*}{dt^{*2}} = -\frac{\mathbf{r}^*}{r^*} \cdot \frac{dU_{cc}^*}{dr^*},\tag{A7}$$

where

$$U_{cc}^{*}(r) = -A\left(\frac{a}{(r^{*2} - (1+a)^{2})} + \frac{a}{(r^{*2} - (1-a)^{2})} + \frac{1}{2}\ln\left[\frac{r^{*2} - (1+a)^{2}}{r^{*2} - (1-a)^{2}}\right]\right),\tag{A8}$$

with

$$A = \frac{\pi^3 \rho_1 \rho_2 \epsilon \sigma^6}{6kT},\tag{A9}$$

and

$$a = R_2/R_1. \tag{A10}$$

The collision rate enhancement factor is:

$$W = \frac{\int_0^\infty \pi b_c^{*2} \nu^* f(\nu^*) d\nu^*}{(1+a)^2},$$
 (A11)

where

$$f(v^*)dv^* = \frac{32v^{*2}}{\pi^2} \exp\left(-\frac{4v^{*2}}{\pi}\right)dv^*.$$
 (A12)

Equations A7-A10 suggest that if the dimensionless quantities A and a are the same, then we will get the same equation of motion and that will lead to the same critical impact parameter  $b_c^*$ . Equations A11-A12 suggest that if we have the same  $b_c^*$  and a, then we will get the same enhancement factor W.

In conclusion, if we have the same dimensionless energy that characterizes the strength of the cluster-cluster interaction, *A*, and the same ratio of the cluster radii, *a*, we will obtain the same collision rate enhancement factor in the IHS model.

Author contributions. VT carried out the MD collision simulations and the PMF calculations with feedback from BR. HY provided the theoretical background, calculations and interpretation regarding the IHS model. VT, HY and BR analysed the data. VT and HY wrote the first draft of the manuscript. BR and HV helped in planning the research and contributed to writing the manuscript.

Competing interests. The authors declare no competing interest.

*Acknowledgements*. Computational resources were provided by the CSC –IT Center for Science Ltd., Finland. The authors wish to thank the Finnish Computing Competence Infrastructure (FCCI) for supporting this project with computational and data storage resources.

420 *Financial Support.* This research has been supported by the European Research Council (project no. 692891 DAMOCLES), the Academy of Finland flagship programme (grant no. 337549) and Centres of Excellence programme (CoE VILMA).

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
