# Peer review of "Gas-phase collision rate enhancement factors for acid-base clusters up to 2 nm in diameter from atomistic simulation and the interacting hard sphere model"

_EGUsphere, 2025_

## Author Response (AR1)

We thank both referees for their positive and constructive feedback, which we carefully considered in the revised manuscript. In our opinion, all the referee comments have been taken into account in a satisfactory, as listed in detail below.

**RC1**

**Minor comments**

1. Throughout the paper  $[HSO_4\cdot (CH_3)_2NH_2\cdot]$  is sometimes referred to as a monomer. I understand that this is because it is treated as the "monomer" in the IHS Hamaker approach, but this terminology can easily lead to confusion. In the field of atmospheric science, this species is typically defined as a dimer. Calling it a "dimer unit" would clarify the meaning without requiring extensive changes to the surrounding text. Alternatively, the authors could explicitly state this naming convention early in the introduction.

We agree that the concept of a "monomer" might be misunderstood by some readers. Therefore, in the revised text the meaning of a monomer is given explicitly: "In this work, the term `monomer' refers to the acid-base dimer unit,  $[HSO_4^-(CH_3)_2NH_2^+]_7$ , because this `heterodimer' is the logical unit in modeling and simulating neutral bisulphate dimethylammonium clusters."

2. For larger SA-DMA clusters, all recent quantum mechanical calculations indicate that clusters with complete proton transfer yield the lowest energy. So I do not think it is a "questionable" assumption at all.

The comment about full proton transfer being "questionable" for these systems has been removed, as suggested, and the sentence modfied:

"For clusters N > 4, we have also assumed complete proton transfer, i.e., clusters consisting only of bisulphate and dimethylammonium ions, in line with the QM minimum energy structures of the largest clusters available in the database (Elm, 2019)."

3. Can the authors explain why the constraint on the radius of gyration was applied? Is the dimer unit unstable with the specific force-field used?

The dimer unit of bisulphate + dimethylammonium is indeed very stable in the OPLS-AA forcefield description. We have clarified the reasoning behind constraining the radius of gyration as follows: "The radii of gyration of each  $[HSO_4\cdot (CH_3)_2NH_2^+]_N$  monomer were constrained by a harmonic upper wall with a spring constant of  $k = 10 \text{ eV/Å}^2$ , starting at the value of  $R_g = 2.5 \text{ Å}$ , to ensure the dimer units remain intact at intermediate distances, while still allowing for necessary rearrangements upon cluster formation."

Also, we added the following text to the discussion regarding larger clusters (for which this constrain becomes more important): "For these simulations, harmonic upper walls were put on the value of the radii of gyration at distances deemed appropriate to achieve a compromise between avoiding elongation of the clusters leading to coalescence already at

intermediate distances, and too rigidly constraining the clusters to their original spherical geometries."

4. Section 3.2: The section focuses heavily on the "attractive tail"; however, the tail region is not explicitly defined. From the plots, I assume it is defined from the shoulder at the largest *r* value and beyond.

This is correct, the meaning of the tail region is now explicitly defined:

"For N = 1, the PMF exhibits a global minimum at r = 4.05 Å with a well depth of 1.53 eV, and the attractive tail for r > 8 Å is in excellent agreement with the rotationally-averaged dipole-dipole interaction (Eq. 16) using the average dipole moment from simulation (see Table 1), indicating that the long-range attractive interaction between the heterodimers can be effectively modeled as that between two point dipoles."

and

"It must be noted that for systems N > 1 we solely focus on the tails of the PMF curves, i.e. distances at which the PMF increases monotonically to zero, for multiple reasons: [...]"

**Technical comments**

- 1. Page 6 line 146: clusterd → cluster: has been corrected
- 2. Page 11 line 265: Perhaps the authors could specify that the different numbers of MD trajectories were due to computational constraints.

We have modified the sentence:

"We performed ntraj = 500, 500, 200, 200 and 100 individual MD trajectory simulations for collisions between N = 1,2,4,8 and 32 [HSO4 (CH3)2NH2+]N, clusters, respectively, to balance between accuracy and increasing computational cost for larger system sizes."

3. Figure 4: Could the 2RHs values be added to the plot to make it easier to follow?

A dashed grey line has been added at  $r=2R_{HS}$  to all panels in the revised Figure 4.

4. Figure 5, 7, 8, and 9: Add that *N* refers to number of the SA-DMA units:  $[HSO_4^-(CH_3)_2NH_2^+]_{N_2}$  to ensure that the figure are self-explanatory.

We have modified the captions of Figures 5, 7, 8 and 9 accordingly.

**RC2**

1. Page 3, line 64-65, I note that only collision between clusters of the same size are studied. Atmospheric NPF involves clusters of diverse sizes (N=1 to N>100N), the same-size collisions (Ni=Nj) would ignore dominant asymmetric cases, missing critical dynamics (e.g., dipole-induced dipole forces in R>1collisions). potentially underestimating rates. The limitations should be discussed here.

We agree that for the full picture of NPF in the atmosphere, collisions between very diverse molecules and clusters need to be considered. Here, we focus on the same-sized cluster-cluster systems as a continuation to our previous work considering molecule-molecule and molecule-cluster collisions. To the revised text we have added the following clarification:

"While the limitation to same-sized cluster collisions in the MD simulations in this work does not account for the vast majority of asymmetric collisions in real atmospheric processes, it still provides a useful starting point to investigate size-dependent collision rate enhancements, when attractive cluster-cluster interactions are taken into account."

2. Page 5, line 139-140, The processes (numerical techniques) to find the minimum of the function  $\omega(r)$  seems to not clearly Additional description need to be provided, as the function  $\omega(r)$  is the key parameter to calculate the enhancement factor of the IHS model.

The revised text has been expanded to make the procedure easier to follow:

"Consequently, we first determine the minimum of  $\omega(r) = r^2(1 - 2U_{cc}(r)/\mu v^2_0)$  numerically. This numerical minimum is subsequently substituted into Eq. 5, facilitating the determination of to determine the critical impact parameter. The obtained critical impact parameter,  $b^2_c = \omega v$  ( $r_{min}$ ), if  $r_{min} > 2R_{HS}$ , or  $b^2_c = \omega v$  ( $R_{HS}$ ), if  $r_{min} \le 2R_{HS}$ , (11) is then used in Eqs. 6 and 7 to calculate the enhancement factor of the IHS model numerically."

3. Page 7, line 195, the authors mentioned that the value of Rb = 0.2RHS was ultimately used, after they test different buffer values Rb, and found the results were not sensitive to this choice. Still, it not clear for the using the value of Rb = 0.2RHS.

The buffer value Rb was first introduced to avoid missing successful collisions in the analysis of the MD collision trajectories, in case clusters changed shape before collision, which could potentially lead to center of mass distances exceeding the sum of hard-sphere radii in the product cluster. However, the results are not sensitive to this choice, and for typical collisions, the center of mass distances were actually significantly smaller than  $2R_{HS}$ , as can be seen in Figure 8 in section 3.3, which makes this point clearer.

We have added the following sentence to the revised manuscript: "In fact, for typical collisions  $r < 2R_{HS}$  (see Fig. 8a in Sec. 3.3)."